# Study on Shrinkage in Alkali-Activated Slag–Fly Ash Cementitious Materials

**DOI:** 10.3390/ma16113958

**Published:** 2023-05-25

**Authors:** Peng Cui, Yuanyuan Wan, Xuejun Shao, Xinyu Ling, Long Zhao, Yongfan Gong, Chenhui Zhu

**Affiliations:** 1School of Transportation and Civil Engineering, Nantong University, Nantong 226019, China; 2Nantong Construction Engineering Quality Testing Center, Nantong 226019, China; 3Nantong Municipal Facilities Management Department, Nantong 226019, China; 4Departments of Civil Science and Engineering, University of Yangzhou, Yangzhou 225127, China; yfgong@yzu.edu.cn

**Keywords:** low-carbon, alkali slag cementitious material, drying shrinkage, autogenous shrinkage, fly ash, fine sand

## Abstract

Traditional silicate cement materials produce a large amount of CO_2_ during production, making it urgent to seek alternatives. Alkali-activated slag cement is a good substitute, as its production process has low carbon emissions and energy consumption, and it can comprehensively utilize various types of industrial waste residue while possessing superior physical and chemical properties. However, the shrinkage of alkali-activated concrete can be larger than that of traditional silicate concrete. To address this issue, the present study utilized slag powder as the raw material, sodium silicate (water glass) as the alkaline activator, and incorporated fly ash and fine sand to study the dry shrinkage and autogenous shrinkage values of alkali cementitious material under different content. Furthermore, combined with the change trend of pore structure, the impact of their content on the drying shrinkage and autogenous shrinkage of alkali-activated slag cement was discussed. Based on the author’s previous research, it was found that by sacrificing a certain mechanical strength, adding fly ash and fine sand can effectively reduce the drying shrinkage and autogenous shrinkage values of alkali-activated slag cement. The higher the content, the greater the strength loss of the material and the lower the shrinkage value. When the fly ash content was 60%, the drying shrinkage and autogenous shrinkage of the alkali-activated slag cement mortar specimens decreased by about 30% and 24%, respectively. When the fine sand content was 40%, the drying shrinkage and autogenous shrinkage of the alkali-activated slag cement mortar specimens decreased by about 14% and 4%, respectively.

## 1. Introduction

Alkali-activated slag cementitious material is a new ‘green’ building material possessing unique advantageous properties, such as a high early strength and good fire, corrosion, seepage, and frost resistance [1,2,3,4]. Furthermore, it bonds strongly with old concrete. Additionally, it consumes less energy during production compared to conventional cementitious materials. Moreover, it can be produced using industrial waste residues, such as fly ash and steel slag. However, alkali slag cement exhibits cracks at early shrinking, thereby restricting its application and further developments [5,6,7]. Hence, its shrinkage performance, mechanical properties, and compliance with the national standards for the setting time must be considered for its widespread application and commercialization.

The shrinkage of alkali cementitious materials is complex and can be caused by several factors [8,9,10,11,12]. Current studies mainly focus on autogenous and drying shrinkage. Autogenous shrinkage refers to the inherent volume deformation of a cementitious material caused by hydration reactions under sealed conditions (with no moisture exchange with the external environment) [13]. When the cementitious material is hardened with a higher density, resulting in the external moisture being less able to penetrate into its interior, the un-hydrated particles of the cementitious material within it continue to hydrate by consuming the residual moisture in the hardened paste. This process is known as self-desiccation and leads to a decrease in the absolute volume of the hardened cementitious paste, which is the fundamental cause of autogenous shrinkage. Drying shrinkage is caused by the evaporation of the internal capillary, gel and adsorbed water in the saturated slurry when exposed to low relative humidity. The causes of shrinkage can be classified into plastic, chemical, autogenous, drying, temperature drop, and carbonation shrinkage.

Based on research, the autogenous shrinkage of alkali slag cement is smaller than that caused by drying; its proportion increases with age [14]. Neto and Cincotto [15] showed that the autogenous shrinkage of an alkali slag cement with 1.7 modulus and 4.5% water glass (WG) at 21d was 4.5 times higher than that of silicate cement at a similar age. As the modulus increases when WG is used as an exciter, the autogenous shrinkage rate during early hydration increases, resulting in a higher final value. Atis [16] demonstrated that WG had the largest drying shrinkage value when used as an exciter, that is, approximately 3–6 times higher than that of ordinary silicate cement as the modulus increased. Sodium silicate exhibited the next highest value, with sodium carbonate being the smallest. Collins and Sanjayan [17] reported similar conclusions. They found that the drying shrinkage of mixed sodium carbonate and sodium hydroxide-excited alkali slag cement was similar to that of silicate cement in the early stage, and it increased after 56d. Chi [18] reported that the drying shrinkage of water-glass-excited slag/fly ash cement was divided into two stages. A higher rate was recorded before 14d, and a lower rate was noted after 14d. The drying shrinkage gradually decreased with an increase in the content of fly ash admixture. Researchers have proposed a series of methods to reduce the shrinkage of alkali-activated slag cementitious materials, including reducing the dosage of cementitious materials, increasing the dosage of aggregates, and incorporating fibers and other shrinkage-reducing materials [19,20,21]. These approaches have shown some degree of success in reducing the shrinkage of alkali-activated materials, but the issue has not been completely resolved.

The aim of this research is addressing the problem of large shrinkage of alkali-activated slag cementitious materials. To mitigate this issue, the author proposes replacing slag powder with fly ash or fine sand to reduce the shrinkage value. Previous studies have investigated the effects of adding fly ash and fine sand on the physical and mechanical properties of alkali-activated slag cementitious materials [22]. The results showed that when the amount of fly ash does not exceed 50% or the amount of fine sand does not exceed 20%, the effect on the strength of the cementitious material is minimal. However, if the amount of these materials exceeds the specified limits, the strength of the alkali slag cementitious material decreases significantly. Additionally, SEM analysis revealed that the density of the cementitious material decreases with the addition of these materials. The current study focuses on the drying shrinkage and autogenous shrinkage of alkali-activated slag cementitious materials with varying amounts of fly ash and fine sand. By combining the findings from previous studies, the authors aim to analyze the influence of fly ash and fine sand content on the shrinkage behavior of alkali-activated slag cementitious materials.

## 2. Materials and Methods

### 2.1. Materials

#### 2.1.1. Slag Powders

The slag powder used in this experiment was sourced from Hunan Huaxin Xianggang Cement Co., Ltd, Hunan, China. It was gray in color and slightly lighter than fly coal. The chemical composition was analyzed using X-ray fluorescence, and the results are presented in Table 1. The powder was sieved through a 0.08 mm square hole sieve to eliminate any impurities. The sieve retention was 2.6%. The Blaine-specific surface area of the powder was 455 m^2^/kg.

#### 2.1.2. Fly Ash

The fly ash used in this experiment was sourced from the China Resources Lianyuan Power Plant (Hunan, China) and was identified as Type II. It was a gray powder with a Blaine-specific surface area of 370 m^2^/kg. The chemical composition was analyzed using X-ray fluorescence, and the results are presented in Table 2.

#### 2.1.3. Fine Sand

The fine sand used in this study, which was used for replacing a portion of the slag powder, was sourced from Liuhe, Nanjing. After screening and grading, the sand particle size was within the range of 0.08 to 0.63 mm.

#### 2.1.4. WG

WG utilized in this experiment was sourced from the Nanjing Chemical Factory. It has a modulus of 3.28 and solid content of 27.43%. The chemical compositions of WG are listed in Table 3. It was diluted with water and later modified with NaOH (analytically pure AR produced by Shanghai Real Chemical Reagent Co., Ltd., Shanghai, China) to create a dilute WG with a modulus of 1.40. The ‘n’ in Na_2_O·nSiO_2_ of WG represents the molar ratio of silicon dioxide to alkali metal oxide, which is equal to the modulus.

#### 2.1.5. Standard Sand

The sand used in the mortar strength test of the alkali-activated slag cementitious material conformed to the Chinese ISO standard sand. The sand used in the drying and autogenous shrinkage tests of the alkali slag cementitious material is fine standard sand with a particle size range of 0.5 to 1.0 mm, based on GB/17671-1999.

#### 2.1.6. Water Mixing

The mixing water for the experiment was obtained from a tap in the laboratory.

### 2.2. Test Method

#### 2.2.1. Test Scheme and Mix Design

In this experiment, the effect of replacing slag powder with varying quantities of fly ash and fine sand on the drying and autogenous shrinkage of alkali-activated slag cementitious materials was investigated. Drying and autogenous shrinkage tests were performed based on the JC/T603-2004 and ASTMC1698-09 test methods, respectively. The mix proportions are presented in Table 4. On the one hand, the F-series test was performed to explore the effect of varying the amounts of fly ash on the drying and autogenous shrinkage of the alkali slag cementitious material. On the other hand, the S-series test demonstrated the impact of different fine sand quantities on the shrinkage. The A-series test revealed the effect of various amounts of fly ash on the autogenous shrinkage of alkali-activated slag cement paste.

The water content of the alkali slag cement mortar (Table 4) was determined by controlling the fluidity of the mortar between 130 and 140 mm. During the experiment, for a similar fluidity, the water demand in the mixed cementitious system of slag powder and fly ash and that of slag powder and fine sand decreased with an increase in the fly ash and fine sand content. This phenomenon can be due to the following factors: (1) The specific surface area of the fly ash and fine sand is smaller than that of slag powder, leading to a naturally lower water demand in the corresponding system. (2) Fly ash is less reactive than slag powder, whereas fine sand is inactive. This results in a decrease in the amount of cementitious material that participates in the early stage of hydration reactions and water demand under a similar fluidity condition. (3) The spherical particles of the fly ash have a “ball effect” that reduce water demand and shrinkage.

#### 2.2.2. Test Method for Dry Shrinkage Performance

The test method for evaluating the drying shrinkage performance of alkali-activated slag cementitious material mortar is based on the JC/T603-2004 “standard for testing drying shrinkage of cement mortar”. The test involves using a triple mold with internal dimensions of 25 mm × 25 mm × 280 mm, and a comparator to measure the drying shrinkage value. The mortar is composed of a ratio of cementitious material to fine standard sand of 1:2, with the water amount adjusted to achieve a flowability of 130 mm–140 mm. The specimens are cured for 1 day in a curing room at 20 ± 1 °C and a relative humidity of not less than 90%, and then demolded. After being placed in a 5% NaOH solution at a temperature of 20 ± 1 °C for 2 days, the specimens are wiped clean and the initial length (L_0_) is measured. Subsequently, the specimens are placed in a drying shrinkage curing box (temperature at 20 ± 3 °C and relative humidity at 50 ± 4%) and the length at the specified age (L_t_) is measured. The shrinkage rate at a certain age is calculated using Equation (1), and the shrinkage rate value is multiplied by 100 during data processing due to the small data. The comparator and the specimen to be tested are placed in a curing box at 20 °C for 2 h before each measurement. Figure 1 shows a physical diagram of the drying shrinkage test.
(1)$$D_{t}=\frac{L_{0}-L_{t}}{250}\times 100^{}$$
where: *D_t_*—Specifies the dry shrinkage rate of the age specimen, (%);

*L*_0_—The initial length of the specimen, (mm);

*L_t_*—Specifies the length of the age specimen, (mm);

250—Effective length of the specimen, (mm).

#### 2.2.3. Test Method for Autogenous Shrinkage

The autogenous shrinkage test of alkali-activated slag cementitious materials mortar is conducted in accordance with ASTM C1698-09 using a corrugated tube [23] mold to cast the slurry. This is because the axial deformation capacity of the corrugated tube is much greater than the radial deformation, and after casting the slurry it can be assumed that the volume deformation of the cementitious material (including axial and radial) is all converted into axial, that is, the length change of the corrugated tube. This not only enables continuous monitoring of autogenous shrinkage of cementitious materials from the beginning of self-casting, but also effectively avoids the influence of gravity, temperature changes, and mold constraints on the test results. Some studies have suggested that the test results are influenced by the size, material, and molding process of the corrugated tube because the initial structure of the cement slurry has not yet formed and the stiffness is low, starting from 1 h after adding water for molding. However, as the hydration of the cementitious material deepens, the stiffness increases continuously, and the influence of the mold stiffness on the test results becomes smaller and smaller compared with the stiffness of the cementitious material. After final setting for 10 h, it can be completely ignored, and the deformation of the entire slurry depends entirely on the deformation of the cementitious material [24,25].

This paper uses the length measurement method to determine the autogenous shrinkage of hardened alkali-activated slag cementitious materials mortar. The experimental device is shown in Figure 2. The molded test mold uses a corrugated tube made of polyethylene, and the inner diameter size of the corrugated tube is 33 mm. Each group of specimens consists of 3, and the shrinkage rate is the average of 3 specimens. After sealing both ends of the corrugated tube, it is horizontally fixed on the rack. The rack is composed of two parallel steel bars with a distance less than the diameter of the corrugated tube. After sealing and fixing both ends, it is in contact with the measuring head of the dial gauge, and the initial length of the specimen is measured after the alkali-activated slag cement has initially set. The reading of the dial gauge is recorded in real-time to obtain the specimen length change value. The shrinkage rate is calculated according to Formula (2):(2)$$A_{t}=\frac{\Delta L}{L}\times 100\%$$
where: *A_t_*—Specifies the autogenous shrinkage rate of the age specimen, (%);

Δ*L*—Change value of specimen length, (mm);

*L*—Original length of specimen, (mm);

## 3. Shrinkage Test Results

### 3.1. Effect of Different Admixtures on the Drying Shrinkage of Alkali-Activated Slag Cementitious Materials

#### 3.1.1. Effect of the Fly Ash Content on the Drying Shrinkage of Alkali-Activated Slag Cementitious Materials

The F-series test was performed based on the proportions listed in Table 4. The samples were removed from the drying box when they reached a specific age, and the experimental data were averaged after discarding outliers. The results are presented in Figure 3.

The drying shrinkage of all of the test pieces increased with aging. The deformation curves of the samples are divided into three distinct stages: rapid shrinkage, and slow and stable development [26]. From 1 to 7d, the specimens were placed in the drying shrinkage curing box. They underwent rapid shrinkage, during which the shrinkage rate increased rapidly. From 7 to 28d, the specimens continued shrinking and deforming. The shrinkage rate gradually decreased with age. Within 28d, the shrinkage of the specimens was almost completed. After this point, the shrinkage rate stabilized and decreased, thereby entering a stable shrinkage-reduction stage. After 56d, the shrinkage rate remained stable with negligible changes. The drying shrinkage of the mortar samples with 100% slag powder increased rapidly, with shrinkage during the first 7d accounting for over 60% of the total shrinkage. After the slag powder was replaced with fly ash, the drying shrinkage of the alkali slag cement mortar specimens decreased as the fly ash content increased. When the fly ash content was 40%, 50%, and 60%, the 28d drying shrinkage of the mortar samples was 0.341%, 0.313%, and 0.281%, respectively. This is approximately 15%, 20%, and 30% lower than the 28 d drying shrinkage of the mortar sample, with a similar aged cementitious system of pure slag powder. The test results show that the replacement of slag powder with fly ash reduces the drying shrinkage of alkali slag cement mortar. The higher the fly ash content, the lower the shrinkage rate. Compared with previous research results, the trend of change in drying shrinkage is consistent, indicating that the addition of fly ash can indeed reduce the drying shrinkage of alkali-activated slag cementitious materials to a certain extent [27]. The influence of fly ash on drying shrinkage can be attributed to the following two aspects:

On the one hand, under conditions of an alkaline solution of WG, the hydrolysis of WG produces hydroxyl ions. This increases the pH of the slurry and dissolves Ca, Mg, and Al in the silica-poor phase of the slag into active cations. As hydration progressed, the silica-rich phase of the slag began hydrating. The active cations and Si ions formed gel hydration products. Simultaneously, Si(OH)_4_ generated by the hydration of WG acts as an active silica phase, thereby reacting with Ca(OH)_2_ to form a C–S–H gel. At the beginning of the hydration, the reaction of the alkali-activated slag progressed rapidly, and the amount of gel products increased. The slurry primarily comprised gel. In a low-humidity environment, the gel hydration product in the slurry changes from a saturated state to an unsaturated state because of the water evaporation in the capillary and gel pores. This leads to the formation of a meniscus in pore water and the shrinkage of the hardened cement paste under the influence of a negative pressure. As the curing age increased, the degree of hydration increased; the structure of the hardened cement paste became more stable, and the volume changes were more moderate.

Contrarily, in a mixed system of alkali-activated slag and fly ash, after fly ash replaces a portion of the slag powder, the hydration reaction of the alkali-activated slag powder first occurs at the initial stage of the reaction. Here, fly ash participation is negligible in the reaction. The pozzolanic reaction is minimal, leading to a decrease in the number of gel products and corresponding shrinkage during drying [5,28]. Additionally, the small-sized particles of fly ash effectively fill and refine the pores of the mortar system, reduce the number of interconnected pores, block connected drainage channels, and hinder water evaporation in the system. The fine aggregate effect of fly ash enhanced the resistance of the mortar to shrinkage deformation and reduced the drying shrinkage to a certain extent.

#### 3.1.2. Effect of the Fine Sand Content on the Drying Shrinkage of Alkali-Activated Slag Cementitious Material

S-series tests were conducted based on the proportions provided in Table 4. The formed samples were removed from the drying box after reaching the required age, and the experimental results were calculated by averaging the data within the error range. The results are shown in Figure 4.

The drying shrinkage of all of the mortar specimens increased as the aging progressed. The shrinkage curves of the specimens are divided into three stages. From 1 to 7d, the specimens were maintained in the drying shrinkage curing box, wherein a rapid increase in the shrinkage rate was recorded. From 7 to 28d, the specimens continued to undergo shrinkage deformation, and the curve was less steep as compared to that before 7d. The upward trend gradually increased with an increase in age. At 28d, the shrinkage of the specimens was almost completed, and they entered a steady shrinkage-reduction stage. At 56d, the shrinkage rate stabilized and remained unchanged. By incorporating fine sand to partially replace the slag powder, the drying shrinkage of the mortar decreased at different ages. With an increase in the fine sand content, the reduction in shrinkage gradually increased. At the age of 7 days, it is evident that with the increase in fine sand content, the drying shrinkage of samples S1–S6 shows a decreasing trend. Among them, the S6 specimen with a fine sand content of 40% has the lowest drying shrinkage. At 28d, the difference in shrinkage between samples with different fine sand contents was further amplified. The shrinkage of the sample containing 40% fine sand was approximately 14% lower than that with 5%. At 56d, the volume of each sample stabilized and showed negligible changes.

This phenomenon can be attributed to the rapid hydration reaction of the sodium silicate-activated slag powder in a sodium silicate alkaline solution. At the initial hydration stage, the number of gel products increased rapidly, and the slurry comprised mainly gels. After soaking in a 5% NaOH solution for 2d, the specimens were transferred to a dry shrinkage curing environment with 50% humidity. Under low-humidity conditions, the gel hydration products in the slurry shrank owing to the water evaporation from the pores and gel pores. As the curing age increased, the hydration process continued. The structure of the hardened cement paste became more stable, and the volume change decelerated. Further, fine sand did not participate in the hydration reaction during the hydration of the slag powder activated by sodium silicate. As the fine sand content increased, the amount of cementitious material in the slurry decreased. This led to a decrease in the total number of gel products generated by the hydration reaction and gradual reduction in moisture loss, thereby slowing down the drying shrinkage. The fine sand in the mortar also contributes to the shrinkage deformation resistance.

### 3.2. Effect of Different Admixtures on the Autogenous Shrinkage of Alkali-Activated Slag Cementitious Materials

#### 3.2.1. Effect of the Fly Ash Content on the Autogenous Shrinkage of Alkali-Activated Slag Cement Mortar

An F-series test was conducted, as outlined in Table 4. The experimental results were calculated by averaging the data within the error range (Figure 5).

The deformation curve of the sample presented in Figure 5 is divided into two stages with distinct change rates in the auto-shrinkage rate. The autogenous shrinkage of all the specimens increased with age. The majority of the increase occurred within the first 10 h, after which the curve plateaued. The addition of fly ash to alkali slag cement mortar resulted in a decrease in the autogenous shrinkage rate. With 20% and 80% fly ash content, the autogenous shrinkage of the mortar was 1730 and 1650 × 10^−6^ mm/mm, respectively. This is 21–24% lower than that of the pure slag powder mortar of a similar age. The results show that increasing the fly ash content reduces the autogenous shrinkage of alkali slag cement mortars.

One of the reasons for this phenomenon is the volcanic ash effect of fly ash. During the initial stage of hydration, the slag powder in the cementitious system reacts with water glass to activate the system, while the hydration reaction of fly ash is slower and does not play a significant role in the hydration process. The addition of fly ash reduces the degree of hydration of the cementitious system, resulting in a decrease in the amount of C-S-H gel generated and a reduction in the amount of water consumed during the hydration reaction. This slows down the rate of water reduction in the slurry, which reduces the negative pressure caused by the curved meniscus of the pore water in the slurry on the cement paste. This, in turn, improves the self-shrinkage phenomenon of the alkali-activated slag cementitious material. Another reason for this is the micro-aggregate effect of fly ash. The particles of fly ash fill the cracks and pores in the hardened slurry, which improves the pore structure [29]. Additionally, the hydration products and its densities of fly ash and slag powder are different. This contributes to the gradual decrease in the self-shrinkage of alkali-activated slag–fly ash cement mortar with increasing fly ash content [30].

Autogenous shrinkage is similar to that of drying shrinkage, which is caused by capillary stress. On the one hand, drying shrinkage in hardened cement slurry is owing to water diffusion into the environment. On the other hand, autogenous shrinkage results from the utilization of internal water during hydration reactions. The reduction in relative humidity in hardened cement slurry occurs by a different mechanism [31].

#### 3.2.2. Effect of the Fine Sand Content on the Autogenous Shrinkage of Alkali-Activated Slag Cement Mortar

The S-series test was conducted as outlined in Table 4. The experimental results were calculated by averaging the data within the error range and are presented in Figure 6.

Figure 6 shows a deformation curve that is divided into two stages with distinct autogenous shrinkage change rates. The shrinkage of all the specimens increased with age. A significant increase was recorded in the first 10 h, and an approximately constant change rate was noted in the subsequent hydration period. By replacing the slag powder with fine sand, the autogenous shrinkage rate of the mortar specimen decreased with an increase in the fine sand content. At 5% and 40% fine sand contents, the autogenous shrinkage rates of the mortar specimens after 11 h were 2155 × 10^−6^ and 1709 × 10^−6^ mm/mm, respectively. Comparatively, the autogenous shrinkage of the mortar specimen using pure slag powder as the binder is 2237 × 10^−6^ mm/mm. These results indicate that replacing slag powder with fine sand reduces the autogenous shrinkage of alkali slag cement mortar. Further, a higher fine sand content results in a lower shrinkage value.

During the early hydration stage, the fine sand exhibited negligible activity and did not participate in the hydration reaction. The system mainly comprised WG, which stimulated the hydration reaction of the slag powder. The introduction of fine sand reduced the amount of colloidal material participating in the hydration reaction. This resulted in a reduction in the amount of the generated C–S–H gel and water consumed during the hydration reaction. Consequently, the water reduction rate in the slurry became smoother, and the negative pressure caused by the hardened cement paste on the pore water tension surface decreased. The self-shrinkage of alkali-activated slag colloidal materials improved. Additionally, fine sand filled the large pores, refined the pore diameter, and improved the pore structure. Further, it can resist shrinkage deformation to a certain extent. Hence, the self-shrinkage of the alkali-activated slag cement mortar gradually decreased with an increase in the fine sand content.

#### 3.2.3. Effect of the Fly Ash Content on the Autogenous Shrinkage of Alkali-Activated Slag Cement Paste

A series of tests were conducted, as presented in Table 4. The experimental results were calculated by averaging the data within the error range and are presented in Figure 7.

The deformation curve of the sample was divided into two stages, with distinct differences in the change rate of the autogenous shrinkage. The autogenous shrinkage of all the specimens increased with age, with rapid growth occurring within the first 10 h, and the growth rate plateaued after 60 h. In the first 10 h, the specimens with 80% fly ash content exhibited an autogenous shrinkage of 1618 × 10^−6^ mm/mm, which was 68% of that observed after 60 h. The autogenous shrinkage rate decreased with an increase in the fly ash content for over 60 h. Here, the autogenous shrinkage rate of the specimens with 80% fly ash content was 58% of that of the pure slag powder specimen with a similar cementitious system. When comparing the data of the systems with 80% and 100% slag powders, the shrinkage rate within the first 20 h was lower than that of the other groups. This may be due to the joint effects of chemical shrinkage and autogenous shrinkage caused by the hydration reaction of the samples, which consumes water from the internal pores.

The reduction in autogenous shrinkage in alkali-activated slag cement paste owing to the addition of fly ash can be attributed to two factors: (1) Fly ash has a lower reaction activity and slower reaction time compared to slag powder. This results in a smaller hydration degree, fewer hydration products, and the reduced shrinkage of the hardened paste. (2) The fly ash particles are small with a large specific surface area. This enables the filling of the pores and improves the pore structure of the hardened cement paste.

## 4. Pore Structure

### 4.1. Test Methods

From the research results above, it can be seen that the use of fly ash and fine sand instead of slag powder can reduce the shrinkage value to a certain extent, and the larger the content, the more the shrinkage value decreases. In order to further study the influence of fly ash and fine sand on the properties of alkali-activated slag materials, combined with the author’s previous research on their mechanical properties, the use of fly ash content exceeding 50% and fine sand content exceeding 20% leads to a rapid decrease in the strength of alkali-activated slag materials. In this paper, paste with a fly ash content of 50% (A4) and mortar with a fly ash content of 50% (F4), as well as mortar with a fine sand content of 20% (S4) were selected to analyze the mercury intrusion porosity (MIP) at 3d, 28d, and 90d, and compared with the paste (A1) and mortar (F1) specimens with 100% slag powder content. The influence of these two materials on the pore structure of alkali-activated slag materials is summarized, and their shrinkage-reduction mechanisms are analyzed. The experimental method for MIP analysis is as follows:

Based on the mix proportions (Table 4), alkali-activated slag cement paste and mortar specimens were prepared and cured in the standard curing room for the specified ages (3d, 28d, and 90d) for mechanical testing. The bean-shaped uncarbonated particles with a diameter of about 5 mm inside the specimens were taken and soaked in anhydrous ethanol solution for 24 h to terminate their hydration before MIP testing. Prior to the testing, the particles were dried to a constant weight in a 60 °C vacuum drying oven. Then, MIP testing was conducted.

### 4.2. Test Results

Figure 8a,b show the pore size distribution of alkali-activated slag materials containing 100% slag powder (A1) and 50% slag powder + 50% fly ash (A4) in the paste at 3d, 28d, and 90d, respectively. Figure 9a–c show the pore size distribution of alkali-activated slag materials containing 100% slag powder (F1), 50% slag powder + 50% fly ash (F4), and 80% slag powder + 20% fine sand (S4) in the mortar at 3d, 28d, and 90d, respectively. The pore structure characteristic parameters are listed in Table 5.

From the results of the MIP analysis, it is evident that the addition of fly ash to the alkali-activated slag specimens leads to an increase in the total porosity of the specimens when compared to the specimens made with pure slag as the cementitious material at the same age. This is one of the primary reasons for the decrease in the strength of the specimens, which is consistent with the author’s previous findings. Moreover, from the perspective of pore size distribution, the proportion of harmful pores with a diameter greater than 1 µm in the specimens with fly ash is significantly reduced. At 28d and 90d, the proportion of harmful pores in the paste specimens is reduced by 66% and 48%, respectively, compared to the pure slag specimens, while in the mortar specimens, the proportion is reduced by 36% and 39%, respectively. On the other hand, the proportion of harmless pores with a diameter less than 0.05 µm increases slightly, with an increase of 23% and 9% in the paste specimens and an increase of 16% and 25% in the mortar specimens at 28d and 90d, respectively. Combined with the changes in the dry shrinkage and autogenous shrinkage values, the presence of a higher proportion of tiny pores is one of the reasons for the reduction in the shrinkage value of the alkali-activated cementitious materials [32]. Additionally, the slower development of strength in the specimens with fly ash is another reason for the reduction in the shrinkage value of these materials.

Compared with pure slag powder mortar specimens of the same age, the addition of fine sand reduces the total porosity, with decreases of 10% and 24% at 28d and 90d, respectively. At 28d, the proportion of harmful pores with a diameter greater than 1 µm was higher in specimens containing fine sand, while the proportion of harmless pores with a diameter less than 0.05 µm was lower. At 90d, the proportion of harmful pores with a diameter greater than 1 µm decreased sharply in specimens containing fine sand, far less than in pure slag powder specimens, and the proportion of harmless pores with a diameter less than 0.05 µm increased slightly. This is because the alkali activity of fine sand is much lower than that of slag powder and fly ash, and it only plays a role in filling pores in the early stage of hydration. In the later stage of hydration, the produced hydration products fill the large-diameter pores in the specimens. In addition, the reduction in the total amount of active cementitious materials results in a decrease in both drying shrinkage and autogenous shrinkage values.

In summary, to reduce the shrinkage of alkali-activated cementitious materials, it is recommended to increase the amount of fine sand appropriately (not exceeding 20%) and reduce the amount of alkali cementitious materials used. Alternatively, a certain amount of fly ash (not exceeding 50%) can be used to replace slag powder to optimize the pore structure. Combined with previous research results, it can be seen that the incorporation of these two materials can effectively improve the pore structure of alkali-activated slag cementitious materials, reduce the proportion of large pores (>1 μm) that are easy to produce volume shrinkage, and increase a large number of small pores (<0.1 μm) that can resist volume shrinkage, which is one of the main reasons for the shrinkage reduction of alkali-activated slag cementitious materials [33,34,35].

## 5. Discussion

This study investigates the drying and autogenous shrinkage values of alkali-activated slag (AAS) cementitious materials with varying fly ash and fine sand contents. With the author’s previous research results on the strength development law of alkali slag cementing materials, it can be found that the addition of these materials can improve the shrinkage performance of AAS at the expense of some mechanical strength [22]. 

In addition, it can be seen from the test results that the autogenous shrinkage is lower than the drying shrinkage for the mixtures with fly ash and fine sand substitution. This is because in the drying shrinkage test, in addition to the volume reduction caused by drying and dehydration, the sample also experienced the self-desiccation process caused by internal hydration, so the measured drying shrinkage value includes its autogenous shrinkage.

## 6. Conclusions

This study investigates the impact of fly ash and fine sand on the drying and autogenous shrinkage of alkali-activated slag cementitious materials when used to replace slag powder. Combined with previous research on its strength development, the following conclusions were drawn:

The addition of fly ash and fine sand reduces the water demand of alkali slag cement mortar, thereby maintaining a similar fluidity.

By replacing slag powder with fly ash, the drying shrinkage of the alkali-activated slag binder materials decreased. The higher the fly ash content, the lower the shrinkage rate. Most of the shrinkage occurred within the first 28d, followed by its gradual deceleration. After 56d, the shrinkage rate stabilized with minimal fluctuations. Furthermore, replacing slag powder with fine sand reduces the drying shrinkage. This is similar to the trend observed for fly ash, i.e., a reduction in shrinkage with increasing fine sand content. 

Replacing slag powder with fly ash reduces autogenous shrinkage. A reduction in shrinkage was observed as the fly ash content increased. The shrinkage rate increased mainly during the first 10 h and thereafter remained stable. Replacing slag powder with fine sand reduces the autogenous shrinkage, exhibiting a trend similar to that observed for fly ash, i.e., a reduction in shrinkage with increasing fine sand content. The development of autogenous shrinkage followed a pattern similar to that of the drying shrinkage.

When the fly ash content is 50%, the 28-day drying shrinkage rate of alkali-activated slag cement mortar decreases by about 20%, and the 10-h autogenous shrinkage rate decreases by about 22%. When the fine sand content is 20%, the 28-day drying shrinkage rate of alkali-activated slag cement mortar decreases by about 9%, and the 10-h autogenous shrinkage rate decreases by about 21%.

Based on the author’s previous research, it was found that by sacrificing a certain mechanical strength, adding fly ash and fine sand can effectively reduce the drying shrinkage and autogenous shrinkage values of alkali-activated slag cement. The higher the content, the greater the strength loss of the material and the lower the shrinkage value.

Moreover, higher contents result in greater strength loss and lower shrinkage values. Future work will focus on the durability of AAS and the impact of these admixtures on its full life-cycle performance. However, despite the addition of these materials, the problem of excessive shrinkage in AAS cementitious materials remains unresolved, with its shrinkage value still higher than that of ordinary Portland cement materials.

## Figures and Tables

**Figure 1 materials-16-03958-f001:**
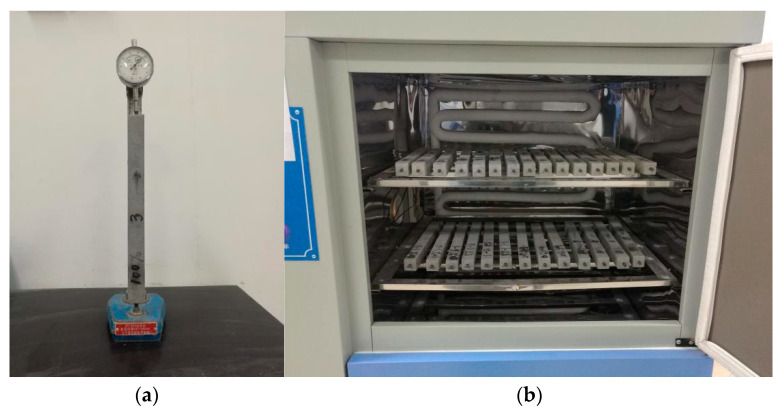
The physical diagram of the drying shrinkage test. (**a**) Comparator, (**b**) Curing in drying shrinkage curing box.

**Figure 2 materials-16-03958-f002:**
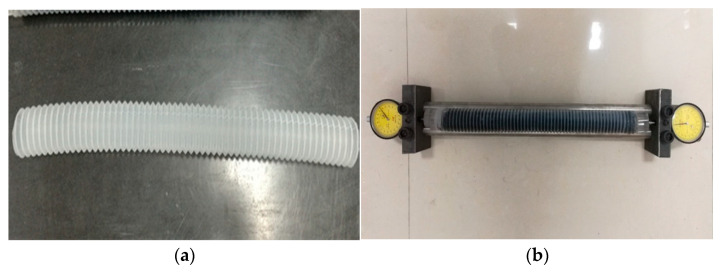
The physical diagram of the autogenous shrinkage test. (**a**) Corrugated tube, (**b**) Measurement diagram.

**Figure 3 materials-16-03958-f003:**
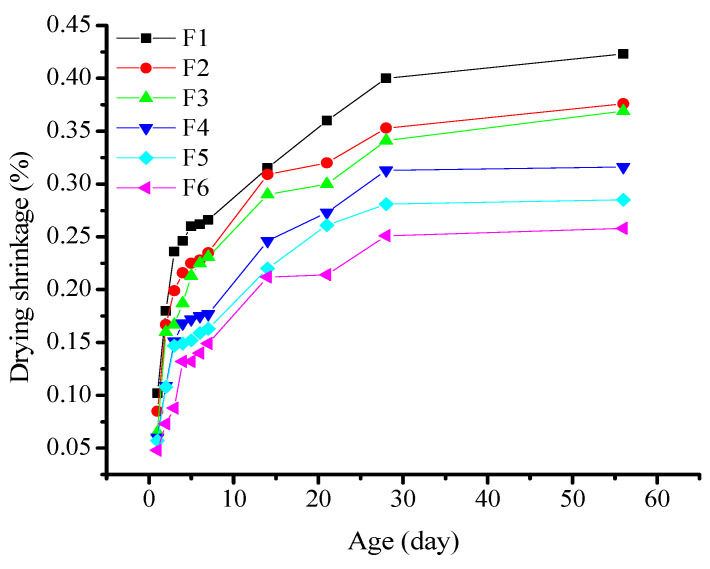
Effect of the fly ash content on the drying shrinkage of alkali-activated slag cementitious materials.

**Figure 4 materials-16-03958-f004:**
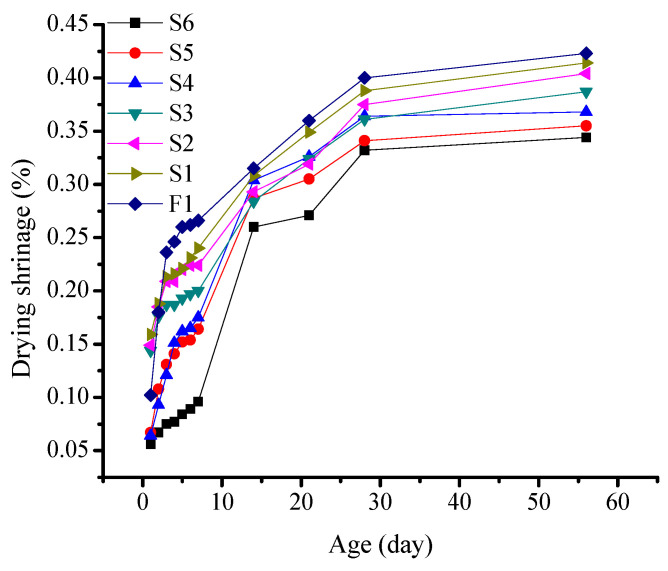
Effect of the fine sand content on the drying shrinkage of alkali-activated slag cementitious material.

**Figure 5 materials-16-03958-f005:**
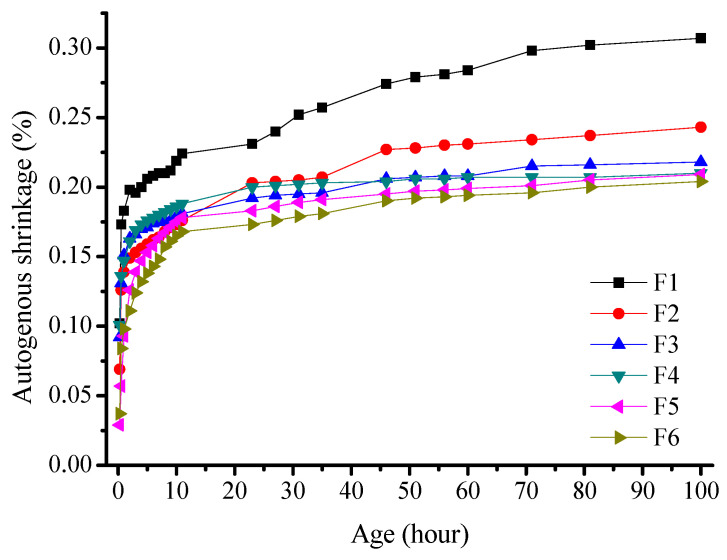
Effect of the fly ash content on the autogenous shrinkage of alkali-activated slag cement mortar.

**Figure 6 materials-16-03958-f006:**
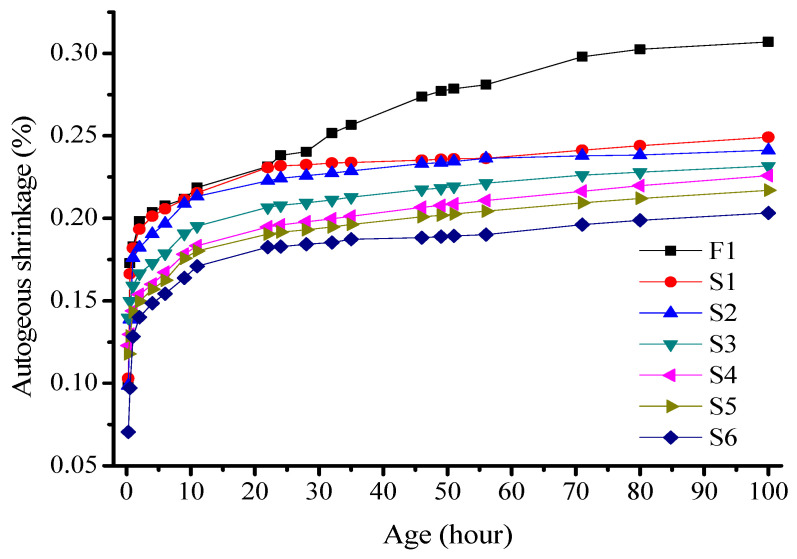
Effect of the fine sand content on the autogenous shrinkage of alkali-activated slag cement mortar.

**Figure 7 materials-16-03958-f007:**
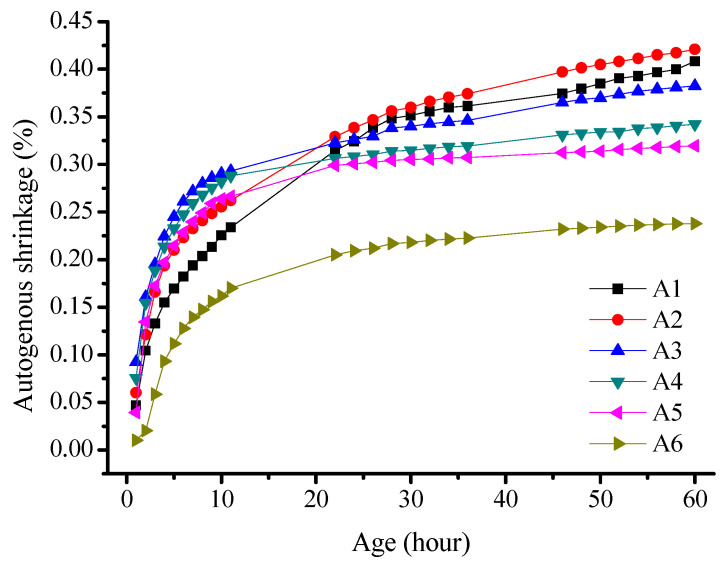
Effect of the fly ash content on the autogenous shrinkage of alkali-activated slag cement paste.

**Figure 8 materials-16-03958-f008:**
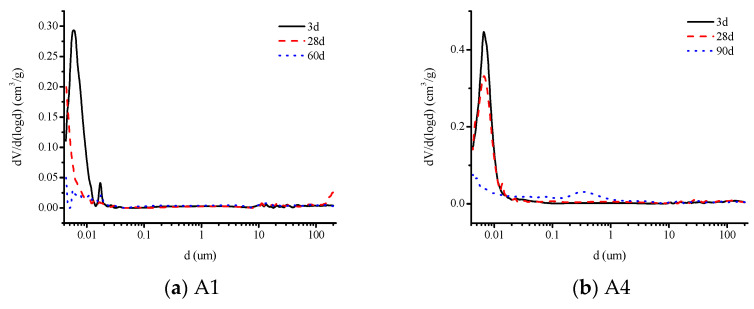
The pore size distribution of alkali-activated slag cement paste A1 and A4 at 3d, 28d, and 90d.

**Figure 9 materials-16-03958-f009:**
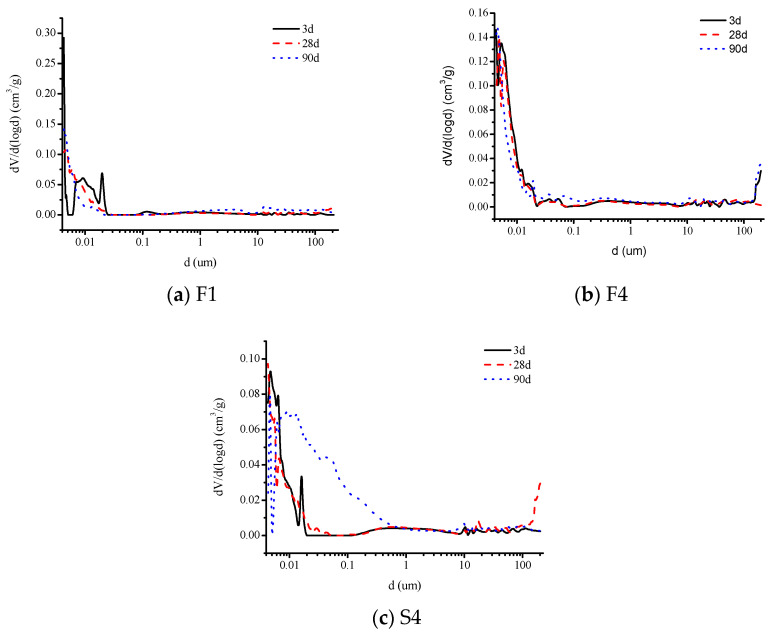
The pore size distribution of alkali-activated slag cement mortar F1, F4 and S4 at 3d, 28d, and 90d.

**Table 1 materials-16-03958-t001:** Chemical constituents of slag powder (wt%).

SiO_2_	Al_2_O_3_	Fe_2_O_3_	CaO	MgO	SO_3_	MnO	TiO_2_	LOI
33.23	17.76	0.416	37.17	7.53	3.10	0.404	0.992	—

**Table 2 materials-16-03958-t002:** Chemical constituents of fly ash (wt%).

SiO_2_	Al_2_O_3_	Fe_2_O_3_	CaO	MgO	SO_3_	Na_2_O	P_2_O_5_	LOI
51.07	36.24	2.88	1.25	0.744	0.394	0.438	0.125	3.72

**Table 3 materials-16-03958-t003:** Chemical constituents of water glass (wt%).

Solid Content	Na_2_O	SiO_2_
36.07	8.64	27.43

**Table 4 materials-16-03958-t004:** Mix ratio of strength experiment (g).

No.	Slag Powder	Fly Ash	Fine Sand	Standard Sand	Water Glass	Water	Total Water
F1	400	0	0	800	136	68	148
F2	320	80	0	800	136	64	144
F3	240	160	0	800	136	60	140
F4	200	200	0	800	136	56	136
F5	160	240	0	800	136	52	132
F6	80	320	0	800	136	48	128
S1	380	0	20	800	129	70	146
S2	360	0	40	800	123	72	143
S3	340	0	60	800	116	73	141
S4	320	0	80	800	109	75	139
S5	280	0	120	800	95	80	136
S6	240	0	160	800	82	86	134
A1	400	0	0	0	136	40	120
A2	320	80	0	0	136	40	120
A3	240	160	0	0	136	40	120
A4	200	200	0	0	136	40	120
A5	160	240	0	0	136	40	120
A6	80	320	0	0	136	40	120

**Table 5 materials-16-03958-t005:** The pore structure characteristic parameters of alkali-activated slag cementitious materials.

No.	Age	Total Porosity (%)	Pore Size Distribution (%)			
			<0.05 µm	0.05 µm~0.1 µm	0.1 µm~1 µm	>1 µm
A1	3 d	17.37	90.59	0.04	2.27	7.10
A4		23.55	90.88	0.80	1.47	6.85
A1	28 d	7.83	68.96	0.40	3.20	27.44
A4		21.32	85.04	1.39	4.23	9.34
A1	90 d	5.02	49.04	2.02	13.14	35.80
A4		12.39	53.25	7.38	20.63	18.74
F1	3 d	8.80	77.34	0.42	9.14	13.1
F4		12.49	76.92	1.1	6.11	15.87
S4		8.11	74.32	0	7.8	17.88
F1	28 d	9.299	75.16	0	3.7	21.14
F4		10.77	78.46	0.92	7.1	13.52
S4		8.33	62.17	0.04	8.42	29.37
F1	90 d	9.79	55.33	0.15	6.5	38.02
F4		12.21	62.02	4.12	10.5	23.36
S4		7.40	65.14	11.49	14.31	9.06

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
