# Peer review of "Study on Shrinkage in Alkali-Activated Slag–Fly Ash Cementitious Materials"

_materials, 2023, doi:10.3390/ma16113958_

Round 1
Reviewer 1 Report
The authors should indicate clearly the conducted tests in the Abstract.
Lines 15-16, revise “However, the shrinkage value during its solidification process is much larger than that 15 of traditional silicate cement materials, severely limiting its promotion and application” to “However, the shrinkage of alkali activated concrete can be larger than that of traditional silicate concrete”
On line 18, sand and fly ash are not admixtures. So, remove “as admixtures”
On lines 43-44, provide a better description of autogenous shrinkage and link to the self-desiccation due to the hydration of cementitious materials in a sealed condition. Use the below reference for more information:
P.-C. Aïtcin, in Science and Technology of Concrete Admixtures, 2016
Before the last paragraph of the introduction, authors should indicate different methods to reduce shrinkage, especially the use of lower cement content, higher aggregate content, the use of fiber and shrinkage reducing materials. The authors can consult with the below references:
Schmitt, T. R.; Darwin, D. Effect of Material Properties on Cracking in Bridge Decks. Journal of Bridge Engineering 1999, 4 (1), 8–13. https://doi.org/10.1061/(ASCE)1084-0702(1999)4:1(8).
Aghaee, K., & Khayat, K. H. (2022). Benefits and drawbacks of using multiple shrinkage mitigating strategies on performance of fiber-reinforced mortar. Cement and Concrete Composites, 133, 104714.
S. a. a. M. Fennis, “Design of ecological concrete by particle packing optimization,” 2011.
Please specify the number of samples in each test.
On line 210, revise “any data outside the error range” to “outliers”
Lines 215, the authors should provide more definition and describe the 3 stages based on the hydration development over time
On line 243 and similar cases, remove “stone”
It is suggested that the authors include compressive strength results to make a conclusion about the final optimum situation.
In the Discussion (Section 5), the authors should describe why the autogenous shrinkage is lower than the drying shrinkage for the mixtures with fly ash and fine sand substitution. Also the higher autogenous shrinkage obtained by the A-series mixtures compared to the F-series and S-series that is due to the lack of sand should be elaborated.
All the comments mentioned in Comments and Suggestions for Authors
Author Response
Dear reviewer,
Thank you very much for your valuable suggestion and for pointing out the shortcomings in the article. I have made the necessary modifications based on your suggestions. If there are any other areas that require further improvement, please do not hesitate to let me know. The detailed corrections are listed below point by point:
- The authors should indicate clearly the conducted tests in the Abstract.
Response 1:Thanks for your comments, the abstract in the manuscript has been revised to indicate the conducted tests.
- Lines 15-16, revise “However, the shrinkage value during its solidification process is much larger than that 15 of traditional silicate cement materials, severely limiting its promotion and application” to “However, the shrinkage of alkali activated concrete can be larger than that of traditional silicate concrete”
Response 2: This part of the manuscript has been revised according to your comments.
- On line 18, sand and fly ash are not admixtures. So, remove “as admixtures”.
Response 3: This part has of the manuscript been revised according to your comments.
- On lines 43-44, provide a better description of autogenous shrinkage and link to the self-desiccation due to the hydration of cementitious materials in a sealed condition. Use the below reference for more information:-C. Aïtcin, in Science and Technology of Concrete Admixtures, 2016.
Response 4: Thanks for your suggestion, this part has been revised according to the reference “P.-C. Aïtcin, in Science and Technology of Concrete Admixtures, 2016”.
- Before the last paragraph of the introduction, authors should indicate different methods to reduce shrinkage, especially the use of lower cement content, higher aggregate content, the use of fiber and shrinkage reducing materials. The authors can consult with the below references:Schmitt, T. R.; Darwin, D. Effect of Material Properties on Cracking in Bridge Decks. Journal of Bridge Engineering 1999, 4 (1), 8- https://doi.org/10.1061/(ASCE)1084-0702(1999)4:1(8). Aghaee, K., & Khayat, K. H. (2022). Benefits and drawbacks of using multiple shrinkage mitigating strategies on performance of fiber-reinforced mortar. Cement and Concrete Composites, 133, 104714. S. a. a. M. Fennis, “Design of ecological concrete by particle packing optimization,” 2011.
Response 5: Thank you for your valuable advice, the introduction has been revised in accordance with your comments.
- Please specify the number of samples in each test.
Response 6: In this experiment, three samples were used for each test. This information is stated in the manuscript.
- On line 210, revise “any data outside the error range”to “outliers”.
Response 7: This part of the manuscript has been revised according to your comments.
- Lines 215, the authors should provide more definition and describe the 3 stages based on the hydration development over time
Response 8: Dear reviewer, the 3 stages in this paper are defined by the author on the basis of the shrinkage curve of alkali-activated slag cementitious materials, and the hydration development at different stages is described in detail in the following text.
- On line 243 and similar cases, remove “stone”.
Response 9: Thanks for your comments, the manuscript has been revised.
- It is suggested that the authors include compressive strength results to make a conclusion about the final optimum situation.
Response 10: Dear reviewer, thank you for your suggestion. Strength results have been included in the conclusion in this paper, and the results of the strength test have been published in “Zhu Chenhui,Wan Yuanyuan,Wang Lei,Ye Yuchen,Yu Houjun,Yang Jie. Strength Characteristics and Microstructure Analysis of Alkali-Activated Slag–Fly Ash Cementitious Material. Materials. 2022,15(17):1-14. DOI: 10.3390/ma15176169”.
- In the Discussion (Section 5), the authors should describe why the autogenous shrinkage is lower than the drying shrinkage for the mixtures with fly ash and fine sand substitution. Also the higher autogenous shrinkage obtained by the A-series mixtures compared to the F-series and S-series that is due to the lack of sand should be elaborated.
Response 11: Thank you for your advice, which is essential to improve the quality of the manuscript. The manuscript has been revised in accordance with your comments. In addition, in the test results, the autogenic shrinkage rate obtained by the A-series mixtures is higher, which is related to the calculation method of autogenic shrinkage rate in this paper(formula 2). Therefore, it was not explained in the discussion, please understand.
Thank you again for your invaluable feedback, and I look forward to receiving your continued guidance in future work.
Best regards,
[Chen-Hui Zhu]

Reviewer 2 Report
It is an interesting work for the scientific community and could be considered for publication in the journal Materials if the proposed comments are resolved.
The summary is well organised, although it lacks a final sentence to close as a conclusion to the research carried out.
References are not formatted in the text, do not use superscripts.
Please eliminate phrases such as line 65: "The author of this study...", use something more impersonal "The aim of this research...".
In the methodology it is necessary to reference the regulations.
The granulometric curve and physical properties of the aggregates used are missing.
Why this fluidity has been chosen as suitable, indicate the standard to which it refers. Include details of the mixing process.
Change Fig.1 to Figure 1, same for all others.
In section 3.1.1. a discussion of the results and comparison with previous studies should be made.
The incorporation of porosimetry is original and supports the results, however the discussion is poor and should be improved. The following references are suggested: https://doi.org/10.3390/s17030522; https://doi.org/10.3989/mc.2018.07817; https://doi.org/10.1016/j.jobe.2020.102097
Include in the conclusions the limitations of the research and future lines of work.
Some expressions throughout the document have a rather informal structure. A thorough reading of the manuscript and a cleaning up of the English language is recommended.
Author Response
Dear reviewer,
Thank you very much for your valuable suggestion and for pointing out the shortcomings in the article. I have made the necessary modifications based on your suggestions. If there are any other areas that require further improvement, please do not hesitate to let me know. The detailed corrections are listed below point by point:
- The summary is well organised, although it lacks a final sentence to close as a conclusion to the research carried out.
Response 1: Thank you for your valuable comments. The summary part of the manuscript has been revised.
- References are not formatted in the text, do not use superscripts.
Response 2: Thank you for reminding me that relevant parts of the manuscript have been revised.
- Please eliminate phrases such as line 65: "The author of this study...", use something more impersonal "The aim of this research...".
Response 3: Thank you for your guidance, humbly accept your advice.
- In the methodology it is necessary to reference the regulations.
Response 4: The test method for evaluating the drying shrinkage performance of alkali-activated slag cementitious material mortar is based on the JC/T603-2004 “standard for testing drying shrinkage of cement mortar”. The test standard ASTM C1698-09 is: Standard Test Method for Autogenous Strain of Cement Paste and Mortar, and it can be used to measure the autogenous shrinkage of cement paste. But there is no test standard for dry shrinkage of cement paste, so it is not measured in this test. The above contents have been stated in the section 2.2.
- The granulometric curve and physical properties of the aggregates used are missing.
Response 5: Dear reviewer, I am deeply sorry for this situation, The granulometric curve and physical properties of the aggregates has been published in “Zhu Chenhui,Wan Yuanyuan,Wang Lei,Ye Yuchen,Yu Houjun,Yang Jie. Strength Characteristics and Microstructure Analysis of Alkali-Activated Slag–Fly Ash Cementitious Material. Materials. 2022,15(17):1-14. DOI: 10.3390/ma15176169”, so I cannot describe it in detail here.
- Why this fluidity has been chosen as suitable, indicate the standard to which it refers. Include details of the mixing process.
Response 6: The determination method of water content in this paper is based on JC/T603-2004 "Test Method for dry shrinkage of cement mortar". The flowability of cement mortar is one of the important indexes to measure the water demand of cement. Using flowability to control the water content can make the physical properties of cement mortar be measured on an accurate and comparable basis, and the measured strength has a good correlation with the concrete strength, which can reflect the actual use effect.
- Change Fig.1 to Figure 1, same for all others.
Response 7: Thanks for your comments, the manuscript has been revised.
- In section 3.1.1. a discussion of the results and comparison with previous studies should be made.
Response 8: Thanks for your comments, section 3.1.1 has been revised, a discussion of the results and comparison with previous studies has been made.
- The incorporation of porosimetry is original and supports the results, however the discussion is poor and should be improved. The following references are suggested: https://doi.org/10.3390/s17030522; https://doi.org/10.3989/mc.2018.07817; https://doi.org/10.1016/j.jobe.2020.102097.
Response 9: Thank you for your valuable comments. Section 4 has been revised according to your comments.
- Include in the conclusions the limitations of the research and future lines of work.
Response 10: Thank you for your valuable comments. The conclusions has been revised according to your comments.
Thank you again for your invaluable feedback, and I look forward to receiving your continued guidance in future work.
Best regards,
[Chen-Hui Zhu]

Round 2
Reviewer 2 Report
The authors have made all the proposed changes and have significantly improved the quality of the submitted manuscript.
No coments